# LITCAB: LIGHTWEIGHT LANGUAGE MODEL CALIBRATION OVER SHORT- AND LONG-FORM RESPONSES

**Xin Liu, Muhammad Khalifa, Lu Wang**
Computer Science and Engineering
University of Michigan
Ann Arbor, MI
{liuxincs, khalifam, wangluxy}@umich.edu

## ABSTRACT

A model is considered well-calibrated when its probability estimate aligns with the actual likelihood of the output being correct. Calibrating language models (LMs) is crucial, as it plays a vital role in detecting and mitigating hallucinations of LMs as well as building more trustworthy models. However, standard calibration techniques may not be suited for LM calibration. For instance, post-processing methods such as temperature scaling do not reorder the candidate generations. On the other hand, training-based methods require fine-tuning the entire model, which is impractical for LMs of large scale. We present LITCAB, a lightweight calibration mechanism consisting of a single linear layer that takes the input text representation and predicts a bias term, which is then added to the LM output logits. LITCAB improves model calibration by only adding $< 2\%$ of the original model parameters. For evaluation, we construct CAT, a benchmark consisting of eight text generation tasks, covering responses ranging from short phrases to paragraphs. We test LITCAB with Llama2-7B, where it improves calibration across all tasks, reducing the average ECE score by as large as 30%. We further conduct a comprehensive evaluation with multiple popular open-sourced LMs from GPT and LLaMA families, yielding the following key findings: **(i)** Larger models within the same family exhibit better calibration on tasks with short generation tasks, but not necessarily for longer ones. **(ii)** GPT-family models show superior calibration compared to LLaMA, Llama2, and Vicuna models, despite having much fewer parameters. **(iii)** Fine-tuning pretrained model (e.g., LLaMA) with samples of limited purpose (e.g., conversations) may lead to worse calibration, highlighting the importance of fine-tuning setups for calibrating LMs.[1]

## 1 INTRODUCTION

While modern language models (LMs) exhibit impressive performance across various tasks (Brown et al., 2020; OpenAI, 2023), they suffer from hallucination (Lin et al., 2021; Zhang et al., 2023), where they can provide nonfactual responses with high confidence. The issue of hallucination undermines user trust and significantly limits the applicability of LMs to domains that require a high degree of reliability, such as legal, financial, and educational sectors. While eliminating hallucination in LMs altogether is highly challenging, calibrating LMs (Nguyen & O'Connor, 2015) by aligning their confidence with the actual probability of output correctness can certainly help. Specifically, a well-calibrated model enables users to gauge the model's confidence and make informed decisions about whether to trust its outputs. Moreover, hallucinated facts can be filtered out when the confidence level is below a certain threshold.

Previous approaches to calibrating neural models mainly fall into two categories: **(i)** post-processing and **(ii)** training-based methods, as outlined in Figure 1. Post-processing techniques have the advantage of not changing the model weights by directly manipulating the sequence probabilities. Example techniques from this family are temperature scaling (Liang et al., 2018) and Platt scaling (Niculescu-Mizil & Caruana, 2005). Post-processing calibration directly adjusts the sharpness of the

---

[1] Data and code are available at `https://github.com/launchnlp/LitCab`.

output distribution, but they do not alter the *relative* confidence rankings among the outputs. This makes them ineffective if one wishes to filter out incorrect outputs based on a confidence threshold. Training-based methods, on the other hand, update the model weights to produce a boost calibration. This family of methods includes label-smoothing (Szegedy et al., 2016), mix-up (Zhang et al., 2020a), or regularization (Pereyra et al., 2017) to mitigate the model's overconfidence. However, training the entire model, which is commonly done for these methods, is impractical for today's LMs with billions of parameters.

Motivated by making training-based calibration more efficient, we present LITCAB, a lightweight calibration technique for LMs. LITCAB takes as input a sequence of hidden states from the LM's final layer and produces a set of logit biases to adjust the generation confidence. Specifically, LITCAB trains a single linear layer on top of the model's last layer using a contrastive max-margin objective, to maximize the token probabilities for correct generations and lower the likelihood for incorrect ones. As LITCAB can adjust the confidence ranking among outputs, a capability that post-processing methods lack, it offers *greater flexibility*. Moreover, the trainable parameters count of LITCAB is **less than 2%** of the original LM parameters, making it significantly more computationally efficient than standard full-model training-based approaches.

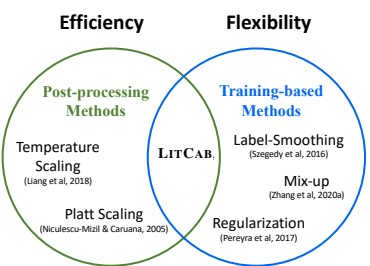

Figure 1: Calibration techniques for neural models. LITCAB combines the benefits of computational efficiency with great flexibility.

We apply LITCAB to Llama2-7B (Touvron et al., 2023b) and compare it against several competitive baselines, including post-processing, training-, verbalization-, and consistency-based methods (Kuhn et al., 2023; Kadavath et al., 2022; Xiong et al., 2023). Our experiments demonstrate the effectiveness of LITCAB, which exhibits uniformly superior calibration than baselines across the text generation benchmarks.

During calibration evaluation, we note that existing work mainly studies *short answer* QA (Tian et al., 2023; Xiong et al., 2023), little attention has been given to LM calibration over *long-form* outputs. To address this gap, we construct and release **Ca**libration evalua**T**ion Benchmark (**CAT**) consisting of *eight* text generation tasks that cover generations encompassing phrases, sentences, and up to paragraphs. We further conduct extensive evaluation over *seven* open-source LMs, including GPT (Radford et al., 2019; Wang & Komatsuzaki, 2021), LLaMA (Touvron et al., 2023a), Llama2 (Touvron et al., 2023b), and Vicuna (Chiang et al., 2023), with sizes ranging from 1.5B to 30B.

Our findings underscore insights regarding the relationships among calibration, model scale, and task performance, as well as the impact of instruction tuning on calibration. First, larger models within the same family demonstrate better calibration for phrase-level tasks, but not necessarily for sentence- and paragraph-level tasks. In addition, when comparing different LM families, despite superior task performance, larger models from various families tend to be worse calibrated compared to the smaller GPT-2 XL (1.5B) model. This observation suggests that model performance and calibration should be optimized concurrently rather than treated as separate objectives. Furthermore, we find that instruction tuning negatively impacts model calibration, as evidenced by Vicuna-13B exhibiting poorer calibration compared to its predecessor, LLaMA-13B.

To summarize, our contributions can be summarized as follows:

- We propose LITCAB, a lightweight calibration mechanism for LMs, which does not need any additional training or fine-tuning of the LM itself and adds $< 2\%$ extra parameters. Experimental results demonstrate its effectiveness compared to methods that use post-processing, model training, verbalization, and self-consistency.

- We construct and release CAT, a benchmark designed for evaluating text generation tasks with responses in both short and long forms. CAT is comprised of eight text generation tasks that include responses ranging from short phrases to sentences and to paragraphs.

- We formulate an evaluation strategy for assessing the calibration of paragraph-level generations. This approach involves extracting distinct claims from the generated content, estimating the confidence associated with each claim, and then analyzing the calibration at the claim level.

- Based on CAT, we conduct an extensive evaluation of the calibration of seven state-of-the-art open-source LMs and extract three key findings.

## 2    RELATED WORK

### 2.1    NEURAL MODEL CALIBRATION

Calibration of neural models was first studied in the context of text classification tasks (Guo et al., 2017; Wenger et al., 2020). Given the goal of strengthening the alignment between model confidence and the likelihood of the output correctness, previous studies can be mainly boiled down to two categories: post-processing and training-based methods. Post-processing methods do not alter the model weights and only adjust the model confidence (i.e., token probabilities). Niculescu-Mizil & Caruana (2005) adopt Platt scaling to calibrate model predicted confidence by fitting a sigmoid function. Similarly, Guo et al. (2017) propose to apply a temperature in the softmax function during model prediction to scale the model predicted probabilities, with the temperature being tuned on the validation data. Zhang et al. (2020a) further improves temperature scaling through ensemble. Post-processing techniques cannot alter the relative rankings among different outputs, thus unsuitable for being directly applied for hallucination mitigation.

As for training-based methods, one common technique is label smoothing (Szegedy et al., 2016), which has been shown to be useful for reducing calibration errors. Alternatively, Pereyra et al. (2017) add a negative entropy penalty to the loss function to diminish overconfidence. Zhang et al. (2020a), on the other hand, regularize the model training by mix-up technique. While training-based methods are generally more flexible than post-editing techniques, full model training is often needed but becomes impractical as LMs grow in size. Instead, LITCAB exhibits the expressivity of training-based methods while eliminating the need for full fine-tuning.

### 2.2    CONFIDENCE ESTIMATION FOR LANGUAGE MODELS (LMS)

An essential step in evaluating the calibration for LMs involves estimating the confidence from the model. Recent research primarily concentrates on methods classified into three types: logit-based, consistency-based, and verbalization-based. Logit-based estimation (Guo et al., 2017; Cheng et al., 2023) measures the model confidence based on the model predicted logits. Consistency-based estimation (Wang et al., 2023; Kuhn et al., 2023) relies on the intuition that LMs will consistently generate similar outputs when they are confident in responding to a query. A major challenge of consistency-based methods is deciding on the semantic equivalence of multiple *long* outputs, which is typically accomplished by utilizing a natural language inference model (Kuhn et al., 2023), BERTScore (Zhang et al., 2020b), or QA-based metrics (Fabbri et al., 2022). However, these methods are limited to sentence-length generations, and it is still unclear how to decide on semantic equivalence over long-form outputs. More recent studies (Tian et al., 2023; Xiong et al., 2023) investigate directly prompting instruction-tuned LMs to verbalize their confidence. While consistency-based and verbalization-based methods have demonstrated their effectiveness in recent LMs, their utilization is largely restricted due to their high computational costs during inference and the requirement for LM's instruction-following capability. In contrast, the LITCAB works only adds minimal computational cost and is suitable for use with any LM whose last-layer hidden states are accessible.

## 3    EVALUATING CALIBRATION FOR GENERATIONS OF VARYING LENGTHS

Model calibration aims to align the model's confidence with the actual likelihood of output correctness. Therefore, one of the key concerns when assessing calibration is model confidence estimation. For classification tasks, this can be achieved by directly using the class probabilities (Guo et al., 2017; Cheng et al., 2023). For generation tasks, however, the output is a sequence of token probabilities, raising the question of how to assess the model's confidence in a generated sequence, e.g., a sentence or a paragraph. Below we describe our methods for eliciting the model confidence for generations of different lengths, as well as evaluating the correctness of model generations.

### 3.1    EVALUATING CALIBRATION FOR GENERATION AT PHRASE- AND SENTENCE LEVEL

When generations are short, e.g., containing one single phrase or sentence, we assume each generation focuses on one idea. Therefore, we aggregate the token-level probabilities into one confidence estimation score at the whole sequence level. Let $x$ denote the input sequence comprising both the in-context learning demonstrations and the question, and let $y$ denote model generation, which consists of $L$ tokens. We determine the corresponding model confidence, denoted as $p(y|x)$, as a

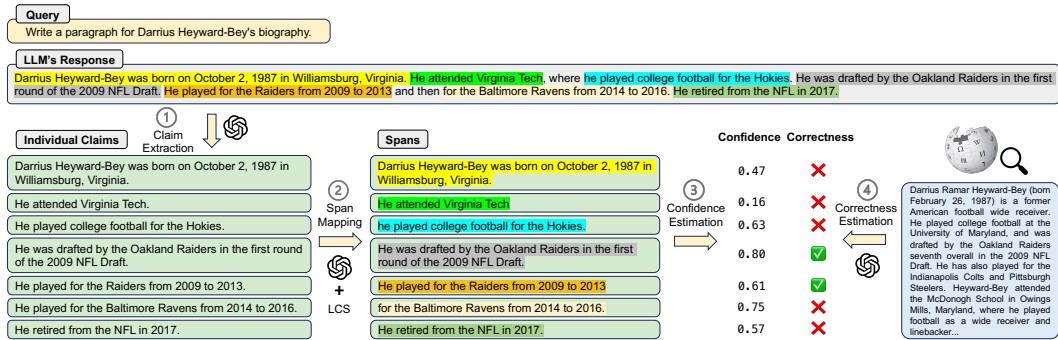

Figure 2: The 4-step process for evaluating calibration over paragraph-level generations by breaking the text down into individual claims and then estimating confidence and judging correctness for each claim separately. **Step 1:** Individual claims are extracted using GPT-3.5-turbo. **Step 2:** The extracted claims are mapped back to the corresponding spans in the paragraph. **Step 3:** The confidence of each claim is estimated by aggregating probabilities over tokens in the corresponding span. **Step 4:** The correctness of each claim is determined by GPT-4 as whether the claim is supported by the retrieved Wikipedia passages.

*geometric* mean over the sequence of token probabilities:

$$p(y|x) = \sqrt[L]{\prod_{t=1}^{L} p(y_t|x, y_{<t})}. \tag{1}$$

Following Tian et al. (2023), we employ GPT-4 to measure the correctness by asking it to decide on the semantic equivalence between the model generations and the references. We consider an output to be correct if there is at least one reference that matches the model output.[2]

## 3.2 EVALUATING CALIBRATION FOR GENERATION AT PARAGRAPH LEVEL

In real-world scenarios, an LM is typically prompted by users to generate long-form text (e.g., one or more paragraphs.) For instance, for a prompt of *"Summarize the events of World War II"*, a good response will need to include multiple claims and facts (henceforth *claims* for simplicity). In such a case, coarse-grained aggregation of the token probabilities over the whole sequence is unsuitable since confidence in individual claims should be estimated separately. As such, it is crucial to break down the model response into distinct claims, treating each as a separate unit for evaluation. In order to do this, we propose a four-step procedure to assess **claim-level confidence** and determine the accuracy of each claim. We demonstrate the evaluation process on a task of generating biography in Figure 2. This procedure is also described below:

- **Step 1: Claim Extraction.** We prompt GPT-3.5-turbo to extract individual claims from the generated text,[3] following Min et al. (2023).

- **Step 2: Span Mapping.** Each extracted claim takes the form of a single sentence. To determine model confidence for each claim using the method in Equation (1), we need to map the claim to the corresponding span in the generated paragraph. To do that, we prompt GPT-3.5-turbo once again to identify the span.[4] Since GPT may rephrase the spans, we select the sentence in the paragraph yielding the longest common string (LCS) with the returned span.

- **Step 3: Confidence Estimation.** Once we gather the generation span that corresponds to each claim, we can calculate the *claim-level confidence* by aggregating the token probabilities of its corresponding generation span, as done in Equation (1).

- **Step 4: Correctness Estimation.** Unlike phrase-level tasks, there is no reference available for evaluating the correctness of claims. Therefore, we follow the retrieval-based method used by Min et al. (2023) to assess the correctness of claims. We first retrieve relevant passages from

---

[2]Prompts for measuring correctness can be found in Appendix A.4.

[3]The prompts used for extracting individual claims can be found in appendix A.1

[4]Prompts can be found in Appendix A.2.

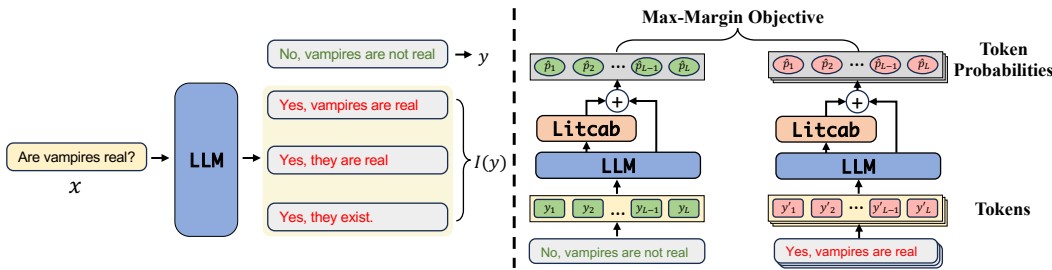

Figure 3: **Left:** The process of constructing positive and negative samples. **Right:** LITCAB Training. LITCAB adjusts the LM predicted logits of the last layer's hidden states, with parameters trained using a max-margin objective.

Wikipedia for each generated claim through a Generalizable T5-based Retriever (Ni et al., 2022).[5] We note that the selection of the information retrieval component does not significantly impact the estimation results, which agrees with Min et al. (2023). We then instruct the more powerful GPT-4 model to determine whether each claim is supported by the retrieved passages. Claims with support are considered correct, otherwise incorrect.[6]

# 4 LITCAB: LIGHTWEIGHT CALIBRATION OF LANGUAGE MODELS

Fine-tuning the LLM to decrease its confidence in incorrect generations and augment its certainty in correct ones has been shown to improve calibration (Jiang et al., 2021). Yet, fine-tuning the full models, especially those of large scale, can be compute-intensive and often require more training samples. In this section, we propose LITCAB for lightweight calibration which trains a *single* linear layer over the LM's last hidden layer representations to predict a bias term, which is used to modify the model's logits and subsequently alter the generation confidence.

Formally, given an LM generation $y$, which consists of $L$ tokens, its last layer hidden states given by the LM can be represented as $\mathbf{h}_{[1:L]} = [\mathbf{h}_1, \mathbf{h}_2, ..., \mathbf{h}_{L-1}, \mathbf{h}_L]$. At each position $t$, LITCAB uses a single linear layer, built on top of model's last layer, to adjust the LM predicted logits $\mathbf{y}_t^{LM} \in \mathbb{R}^V$ by using $\mathbf{y}_t = \mathbf{y}_t^{LM} + \mathbf{W}\mathbf{h}_t + \mathbf{b}$, where $\mathbf{W}$ and $\mathbf{b}$ are the trainable parameters. $V$ denotes the vocabulary size. After that, the adjusted sentence confidence $\hat{p}(y|x)$ can be derived from the geometric mean of the updated token probabilities $\hat{p}(y|x) = \sqrt[L]{\prod_{t=1}^L \hat{p}(y_t|x, y_{<t})}$. Here $\hat{p}(y_t|x, y_{<t})$ is the updated probability of token $y_t$ computed by the softmax function:

$$\hat{p}(y_t|x, y_{<t}) = \frac{\exp \mathbf{y}_t^{(y_t)}}{\sum_{v=1}^V \exp \mathbf{y}_t^{(v)}}, \tag{2}$$

where $\mathbf{y}_t^{(y_t)}$ is the logit corresponding to token $y_t$ in the vector $\mathbf{y}_t$.

**Negative Sample Construction and Max-margin Objective.** We gather a collection of positive and negative samples to train our model for each task. For phrase- and sentence-level tasks where references are available, we use the references directly as positive samples. We then repeatedly sample model generations, treating incorrect ones as negative samples. For the paragraph-level tasks, we directly instruct each LM to generate biographies or descriptions for the given people's names or entities. Subsequently, we categorize the correct claims as positive samples and the incorrect ones as negative samples. Let $x$ represent the input and $y$ represent the positive sample, while $I(y)$ represents the negative samples. The max-margin objective for training the LITCAB is given by $L(x, y) = \sum_{y' \in I(y)} \max(0, 1 + \hat{p}(y'|x) - \hat{p}(y|x))$. Intuitively, the max-margin objective enables the LM to distinguish correct generations from incorrect ones through LITCAB, thereby yielding enhanced calibration of confidence for both correct and incorrect generations. We collect 3 incorrect samples per question for phrase- and sentence-level tasks, and use all generated positive and negative samples for paragraph-level tasks. The process of constructing positive and negative samples and the training process are illustrated in Figure 3.

---

[5]We use `gtr-t5-large` from Huggingface.

[6]Prompt for estimating claim correctness is listed in Appendix A.3.

## 5 CAT: A BENCHMARK FOR CALIBRATION EVALUATING

Unlike previous studies that have primarily concentrated on multiple-choice QA settings (Jiang et al., 2021; Cheng et al., 2023), we emphasize on general text generation tasks with **responses of varying lengths**, which present more challenges to assessing LM calibration. Therefore, for evaluation, we collect tasks with lengths of responses at **(i)** phrase level, **(ii)** sentence level, or **(iii)** paragraph level. For phrase-level generation datasets, we use **NaturalQuestions (NQ)**, **SciQ**, and **TriviaQA**, all of which include short responses (e.g., named entities). For sentence-level responses, we choose **TruthfulQA** and **WikiQA**, where model responses are complete sentences. For paragraph-level generations, we prompt the LMs to write biographies of different figures (celebrities, scientists, etc.), whose names are sourced from **BioGen** (Min et al., 2023). As these names originally come from Wikipedia, we can easily verify the model-generated claims by comparing them against the corresponding Wikipedia passage. We also design a paragraph-level task called **WikiGen** (new), where LMs are tasked to generate Wiki-style descriptions for entities gathered from the fact verification dataset FEVER (Thorne et al., 2018). Compared with BioGen which is limited to biographies, Wiki-Gen covers a wider spectrum of subjects beyond individuals, including events, concepts, and objects, resulting in more diverse topics. Additionally, we employ **QAMPARI** (Amouyal et al., 2022), a question-answering task with multiple answers. While the answers are not long, the model's response comprises multiple claims and can be easily extracted. Instead of searching for a Wikipedia passage to assess correctness, we directly evaluate it by comparing the generated claims with the ground truth. The statistics and examples of the benchmark, along with details on the construction of training and test sets, are listed in the Appendix B.

## 6 EXPERIMENTS

### 6.1 EVALUATION METRICS

We evaluate the model calibration performance using three metrics:

- **Expected Calibration Error (ECE).** Following previous studies (Guo et al., 2017; Lin et al., 2022; Tian et al., 2023), we adopt the expected calibration error, which estimates the gap between the model confidence and the actual accuracy. We divide the confidence from 0 to 1 into 10 intervals with a spacing of 0.1, and assign samples to the interval corresponding to their confidence scores. Within each bin $b_i$, we compute the accuracy of model generations $acc(b_i)$ and the average model confidence $conf(b_i)$. The ECE is then calculated as $ECE = \sum_{i=1}^{K} \frac{|b_i|}{N} |acc(b_i) - conf(b_i)|$, where $N$ denotes the number of model generations. Lower ECE indicates better calibration, reflecting closer alignment between confidence and accuracy.

- **Brier Score** directly measures the distance between the model confidence and the binary correctness label of the generation. Given all model generations $Y$, the Brier score is defined as $\text{Brier} = \frac{1}{N} \sum_{y \in Y} (p_y - \mathbb{I}(y))^2$, where $\mathbb{I}(y)$ denotes the correctness label.

- **Selective Classification Metrics.** Following Tian et al. (2023), we also evaluate calibration by employing metrics from the selective classification domain, as model confidence plays a pivotal role in this field. Specifically, we employ the metrics of **accuracy at coverage** (acc@q) and **coverage at accuracy** (cov@p). Acc@q quantifies the precision of the model by evaluating the accuracy of the top $q$ percent of predictions. Conversely, cov@p gauges the extent to which the model achieves recall by identifying the largest percentage, denoted as $c$, for which the most confident $c$ percent of predictions exhibit accuracy surpassing the threshold $p$. Compared to AUROC (Bradley, 1997) that focuses on measuring the quality of confidence scores, these two metrics directly evaluate the LM's performance in filtering out incorrect generations by setting the thresholds.

### 6.2 BASELINES

We compare LITCAB with two popular calibration methods for neural networks on the widely-used Llama2-7B[7], which is one of the most recent open-source LMs:

---

[7]We choose the 7B version because it can be easily fine-tuned using LoRA, enabling a comparison between LITCAB and training-based methods.

- **Temperature Scaling** (Liang et al., 2018): A temperature constant is used to scale the logits before computing the softmax. We employ gradient descent optimization to search for the optimal temperature on the training set.
- **Label Smoothing** (Szegedy et al., 2016): As a training-based method, the entire LM is fine-tuned on training set with label smoothing. To train Llama2-7B, we adopt LoRA (Hu et al., 2022) to enable the fine-tuning process on a single GPU.

Additionally, we also compare LITCAB with recent confidence estimation methods that are specifically designed for LMs, including:

- **Verbalization** prompts the LLM to provide its own confidence in a given output. We directly reuse the prompt provided in Tian et al. (2023).
- **P(IK)** (Kadavath et al., 2022): A linear layer is stacked on top of the LM last-layer's hidden state that corresponds to the question's last token. The added layer learns to predict whether the model can accurately answer the question. We keep the parameters of the LM fixed and only fine-tune the linear layer.
- **Self-Consistency** (Tian et al., 2023; Xiong et al., 2023) relies on the hypothesis that confident responses will appear more frequently when sampling from the model. As applying majority voting directly is not straightforward over long-form generations, we rely on a natural language inference model. Details about this method are listed in Appendix C.

We note that LM confidence estimation methods, i.e. Verbalization, P(IK) and Self-Consistency, can only yield a single unified score for the entire generation, which cannot be used to assess calibration at the claim level since each generation normally contains multiple claims. Therefore, we do not list the results for paragraph-level tasks by these comparison methods. To construct training instances for label smoothing and temperature paragraph-level tasks, we employ the same generation process used for constructing training instances for LITCAB, but only keep the accurate claims. Other generation and training details (e.g. hyperparameters) can be found in Appendix C.

## 6.3 RESULTS COMPARED WITH TRADITIONAL CALIBRATION METHODS

The results of LITCAB and the two commonly used neural network calibration methods are shown in Table 1. As can be seen, LITCAB consistently improves over the original LM's calibration performance across all tasks, highlighting the effectiveness of LITCAB. Using the NQ dataset, a dataset of human-originated queries via Google search, as an example, LITCAB reduces ECE by 41%, from 0.171 to 0.101. Furthermore, when compared to the two baselines, LITCAB achieves the best calibration performance, as measured by the lowest average ECE (0.093) and Brier score (0.197) across all tasks. Moreover, LITCAB further enhances the model's calibration performance on phrase- and sentence-level tasks when combined with temperature scaling, suggesting its compatibility with other calibration methods. However, the combination of LITCAB with temperature scaling leads to poorer calibration performance on paragraph-level tasks. This occurs because the temperature is adjusted based on the model output samples due to the absence of references, which further intensifies overconfidence.

Note that the fine-tuning of the Llama2-7B model with label smoothing does not yield satisfactory calibration performance, possibly due to the small size of training data that leads to model overfitting. In contrast, LITCAB can leverage small training sets for improved calibration, suggesting that LITCAB achieves superior data efficiency compared to label smoothing.

## 6.4 RESULTS COMPARED WITH LM CONFIDENCE ESTIMATION METHODS

As shown in Table 1, LITCAB outperforms all LM confidence estimation methods in phrase- and paragraph-level tasks and attains the lowest average ECE and Brier score, again showcasing its superior effectiveness in calibrating LM. Interestingly, we can observe that the model's confidence estimated by verbalization is worse than that of the original model for most tasks. We speculate that this could be attributed to the instruction-following skills that verbalization requires, which Llama2-7B lacks. Despite the specific instances where self-consistency surpasses LITCAB in tasks like NQ and TruthfulQA, LITCAB consistently demonstrates competitive performance across a wide range of tasks, as measured by its lower averaged ECE (0.125 vs. 0.084) and Brier score (0.216 vs. 0.195) on

| Task | Metric | | Original LM | Traditional Calibration Methods | | LM Confidence Estimation Methods | | | LITCAB | LITCAB w/ Temp. Scaling |
|------|--------|--|-------------|---------------|--------------|-------|---------------|------------------|--------|------------------------|
| | | | | Label Smoothing | Temp. Scaling | P(IK) | Verbalization | Self-Consistency | | |
| *Phrase Level* | | | | | | | | | | |
| NQ | acc@50 | ↑ | 0.288 | 0.208 | 0.288 | 0.286 | 0.254 | **0.340** | 0.300 | 0.300 |
| | cov@50 | ↑ | 0.115 | 0.061 | 0.115 | 0.000 | 0.055 | **0.217** | 0.105 | 0.105 |
| | ECE | ↓ | 0.171 | 0.186 | 0.165 | 0.158 | 0.516 | 0.145 | 0.101 | **0.083** |
| | Brier | ↓ | 0.196 | 0.212 | 0.193 | 0.204 | 0.468 | 0.163 | 0.169 | **0.164** |
| SciQ | acc@50 | ↑ | **0.764** | 0.212 | **0.764** | 0.656 | 0.660 | 0.744 | 0.762 | 0.762 |
| | cov@90 | ↑ | 0.211 | 0.003 | 0.211 | 0.004 | 0.117 | 0.124 | **0.221** | **0.221** |
| | ECE | ↓ | 0.094 | 0.391 | 0.091 | 0.188 | 0.318 | 0.101 | 0.084 | **0.082** |
| | Brier | ↓ | 0.203 | 0.386 | **0.202** | 0.276 | 0.344 | 0.227 | 0.203 | 0.203 |
| TriviaQA | acc@50 | ↑ | **0.500** | 0.302 | **0.500** | 0.372 | 0.404 | 0.446 | 0.478 | 0.478 |
| | cov@60 | ↑ | 0.111 | 0.019 | 0.111 | 0.023 | 0.053 | 0.079 | **0.201** | **0.201** |
| | ECE | ↓ | 0.112 | 0.184 | **0.079** | 0.215 | 0.431 | 0.181 | 0.081 | **0.079** |
| | Brier | ↓ | 0.203 | 0.259 | **0.195** | 0.277 | 0.409 | 0.253 | 0.203 | 0.199 |
| *Sentence Level* | | | | | | | | | | |
| TruthfulQA | acc@50 | ↑ | 0.314 | 0.181 | 0.314 | 0.267 | 0.233 | **0.405** | 0.314 | 0.314 |
| | cov@40 | ↑ | 0.136 | 0.000 | 0.136 | 0.005 | 0.224 | **0.500** | 0.195 | 0.195 |
| | ECE | ↓ | 0.138 | 0.134 | 0.161 | 0.323 | 0.510 | **0.060** | 0.105 | 0.103 |
| | Brier | ↓ | 0.218 | **0.175** | 0.240 | 0.349 | 0.474 | 0.194 | 0.206 | 0.203 |
| WikiQA | acc@50 | ↑ | 0.388 | 0.273 | 0.388 | 0.339 | 0.372 | **0.628** | 0.397 | 0.397 |
| | cov@50 | ↑ | 0.012 | 0.000 | 0.012 | 0.004 | 0.202 | **0.621** | 0.062 | 0.062 |
| | ECE | ↓ | 0.075 | 0.155 | **0.066** | 0.239 | 0.535 | 0.136 | 0.075 | 0.074 |
| | Brier | ↓ | 0.212 | 0.239 | 0.222 | 0.299 | 0.518 | 0.243 | 0.212 | **0.210** |
| *Paragraph Level* | | | | | | | | | | |
| BioGen | acc@50 | ↑ | 0.347 | 0.334 | 0.347 | – | – | – | **0.354** | **0.354** |
| | cov@40 | ↑ | 0.066 | 0.059 | 0.066 | – | – | – | **0.148** | **0.148** |
| | ECE | ↓ | 0.169 | 0.196 | 0.246 | – | – | – | **0.166** | 0.243 |
| | Brier | ↓ | 0.269 | 0.284 | 0.313 | – | – | – | **0.267** | 0.308 |
| WikiGen | acc@50 | ↑ | **0.876** | 0.860 | 0.876 | – | – | – | 0.872 | 0.872 |
| | cov@80 | ↑ | 0.745 | 0.733 | 0.745 | – | – | – | **0.756** | **0.756** |
| | ECE | ↓ | 0.045 | 0.075 | 0.049 | – | – | – | **0.037** | 0.065 |
| | Brier | ↓ | 0.172 | 0.187 | 0.173 | – | – | – | **0.171** | 0.174 |
| QAMPARI | acc@50 | ↑ | 0.193 | 0.180 | 0.193 | – | – | – | **0.207** | **0.207** |
| | cov@30 | ↑ | **0.260** | 0.156 | **0.260** | – | – | – | 0.257 | 0.257 |
| | ECE | ↓ | 0.290 | 0.213 | 0.303 | – | – | – | **0.096** | 0.104 |
| | Brier | ↓ | 0.228 | 0.208 | 0.273 | – | – | – | **0.142** | 0.157 |
| **Average** | acc@50 | ↑ | 0.459 | 0.319 | 0.459 | – | – | – | **0.461** | **0.461** |
| | ECE | ↓ | 0.137 | 0.191 | 0.145 | – | – | – | **0.093** | 0.104 |
| | Brier | ↓ | 0.213 | 0.245 | 0.226 | – | – | – | **0.197** | 0.203 |

Table 1: Results of LITCAB and baselines on CAT, with the best score per metric per dataset in **bold**. We highlight numbers where LITCAB improves over both the original LM and all baselines in  blue ; when LITCAB outperforms the original LM, it is colored in  green . The last row shows the **average** metric value over all tasks. LITCAB effectively calibrates the LM on various tasks. The combination of LITCAB and Temp. Scaling further enhances the LM's calibration performance on phrase- and sentence-level tasks.

phrase- and sentence-level tasks. This underscores the versatility and effectiveness of our approach, making it a promising choice for various tasks. Additionally, considering that the computational cost of self-consistency during inference is $N$ times[8] higher than that of LITCAB, our approach is computationally more effective.

## 6.5 INVESTIGATING CALIBRATION OF LMs USING CAT

Here on CAT we assess a broad range of popular open-source LMs, with sizes ranging from 1.5B to 30B. Specifically, we evaluate GPT2-XL (1.5B) (Radford et al., 2019), GPT-J (6B) (Wang & Komatsuzaki, 2021), LLaMA families (7B, 13B, and 30B) (Touvron et al., 2023a), Llama2 families (7B and 13B) (Touvron et al., 2023b), and Vicuna-v1.3-13B (Chiang et al., 2023). The results are illustrated in Figure 4.[9] From the figure, we can readily draw the following conclusions:

**Larger models within the same family generally exhibit better calibration on short model responses, but not necessarily for longer ones.** Within the models trained using the same data, i.e., LLaMA families and Llama2 families, the large ones of LLaMA-30B and Llama2-13B can achieve not only improved task performance (acc@q, cov@p) but also superior calibration (ECE, Brier) that their smaller counterparts on phrase-level tasks. This echoes the findings in Kadavath et al. (2022). However, LLaMA-30B does not demonstrate better calibration (ECE, Brier) in tasks

---

[8] $N$ denotes the number of model predictions in self-consistency, which is set as 10 in our setting.

[9] Detailed results can be found in Appendix D

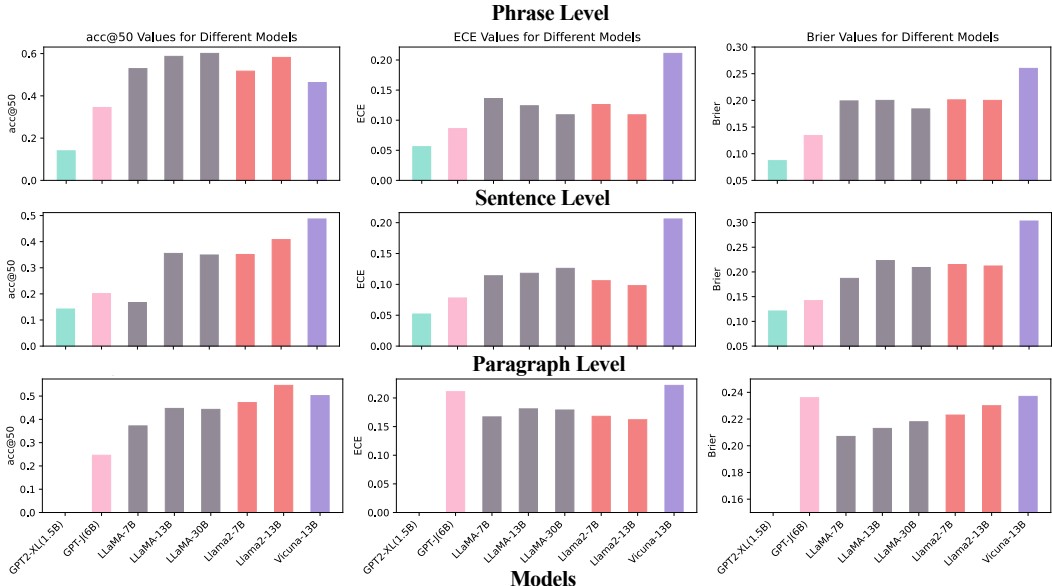

Figure 4: Bar charts of averaged acc@50, ECE, and Brier score of popular LMs computed on CAT. The results for GPT-2 XL in paragraph-level tasks are missing due to the prompt's length exceeding its context limit. Bars with the same color represent models from the same model family.

involving longer generations (sentence- and paragraph-level tasks). This suggests that the model scaling effect may not hold true for calibration in tasks with longer response lengths.

**GPT2-XL is better calibrated than other larger models.** Despite its smaller model size, GPT2-XL exhibits strong calibration, as evinced by its consistently lower ECE and Brier scores in comparison to the larger LMs, including the largest model, LLaMA-30B. However, note that LLaMA models tend to yield better accuracy, as measured by higher acc@q and cov@p scores. We observe that overconfidence is infrequently observed by both GPT2-XL and GPT-J, especially on the incorrect generations, thus leading to better calibration. These observations suggest that model accuracy and calibration should not be treated as separate objectives to optimize. Given their interconnectedness, improving them concurrently seems to be a far more favorable and efficient approach in future research undertakings.

**Vicuna, fine-tuned from LLaMA, gains worse calibration than LLaMA.** Vicuna-13B, which is fine-tuned from LLaMA-13B on user-shared conversations, exhibits a much worse calibration. This is highlighted by the increased average ECE and Brier scores. This suggests caution when implementing a second stage of fine-tuning, as it could potentially diminish calibration.

## 7 CONCLUSION

We presented LITCAB, a lightweight calibration technique for LMs. LITCAB calibrates LMs by predicting a bias term added to the model's predicted logits. An empirical comparison with post-processing, training-, verbalization-, and consistency-based approaches substantiates the efficacy of LITCAB and underscores its computational efficiency, considering that it introduces less than 2% of additional parameters. Moreover, we collected CAT, a benchmark specifically designed to evaluate calibration in text generation settings with both short- and long-form responses. We propose an evaluation methodology dedicated to examining calibration in long-form generations, which can serve as the basis for future studies. Finally, we use CAT to comprehensively assess seven open-source LMs. Our findings show that larger models within the same family generally exhibit superior calibration on phrase-level tasks, but not necessarily on tasks at sentence and paragraph levels. Interestingly, despite a lower accuracy, GPT2-XL (1.5B) is better calibrated than larger models. Our observations also reveal that additional fine-tuning may lead to worse calibration.

**Acknowledgments.** This work is supported in part by National Science Foundation through grant IIS-2046016 and LG AI Research. We also thank ICLR reviewers for their helpful comments.

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

## A  PROMPTS

### A.1  PROMPT FOR BREAKING DOWN PARAGRAPHS

```
Please break down the following paragraph into independent facts.
You should ONLY present the independent facts (one in a row), no
other words or explanation.
```

-> [Biography]

### A.2  PROMPT FOR MAPPING CLAIMS WITH SENTENCE SEGMENTS

```
Which segment in the following paragraph reflects the claim
"[claim]"? The segment doesn't need to be a complete sentence
and should be as short as possible.
```

-> [Biography]

### A.3  PROMPT FOR ASSESSING CORRECTNESS OF CLAIMS

```
Answer the question about [Name] based on the given context.

Title: [Passage Title]

Text: [Passage]

Input: [claim] True or False?

Output:
```

### A.4  PROMPT FOR ASSESSING CORRECTNESS OF MODEL GENERATION

```
Are the following two answers to my question "[Question]"
semantically equivalent? (Answer "Yes" or "No" first, and then
explain your answer.)

1. [Reference]

2. [Model Response]
```

| | NQ | SciQ | TriviaQA | TruthfulQA | WikiQA | BioGen | WikiGen | QAMPARI |
|---|---|---|---|---|---|---|---|---|
| # Train | 2K | 2K | 2K | 397 | 1040 | 500 | 500 | 2K |
| # Test | 1K | 1K | 1K | 420 | 293 | 183 | 100 | 1K |
| Avg. # Response tokens | 2.2 | 1.5 | 2.3 | 35.5 | 25.7 | 77.3 | 104.0 | 26.5 |
| Max. # Response tokens | 6 | 12 | 42 | 24 | 166 | 318 | 197 | 458 |
| Min. # Response tokens | 1 | 1 | 1 | 1 | 1 | 21 | 33 | 3 |

Table 2: Statistics of tasks in CAT. The number of response tokens for phrase- and sentence-level tasks is estimated based on references. For BioGen and WikiGen, the number of response tokens is calculated based on the generations of Llama2-7b, due to the lack of references.

| Task | Example |
|---|---|
| ***Phrase Level*** | |
| NQ | *Question*: what is the smallest prime number that is greater than 30 
 *Response*: 31 |
| SciQ | *Question*: About how tall can mid-ocean ridges be? 
 *Response*: about 2 km |
| TriviaQA | *Question*: In which decade did stereo records first go on sale? 
 *Response*: 1930's |
| ***Sentence Level*** | |
| TruthfulQA | *Question*: Which language do all people in Europe speak? 
 *Response*: There's no single language that all Europeans speak |
| WikiQA | *Question*: how much is 1 tablespoon of water 
 *Response*: In Australia one tablespoon (measurement unit) is 20 mL. |
| ***Paragraph Level*** | |
| BioGen | *Question*: Write a paragraph for Bill Tobin's biography. 
 *Response*: Ron Meagher (born October 2, 1941, Oakland, California, USA) is best known as the bassist of the American rock band The Beau Brummels. When guitarist-songwriter Ron Elliott was putting the... |
| WikiGen | *Question*: Write a paragraph about The Beatles. 
 *Response*: The Beatles were an English rock band formed in Liverpool in 1960, comprising John Lennon, Paul McCartney, George Harrison, and Ringo Starr. They are regarded as the most influential band of all time... |
| QAMIPARI | *Question*: What fictional character had their debut in Mega Man X? 
 *Response*: Flame Mammoth; Spark Mandrill; Launch Octopus; Chill Penguin; Sigma; Storm Eagle; Zero; Boomer Kuwanger; Sting Chameleon; Armored Armadillo |

Table 3: Examples of question response pairs in the benchmark.

## B  DETAILS OF CAT

We list the statistics of tasks from CAT in Table 2 and show some examples of CAT in Table 3. For phrase- and sentence-level tasks, we randomly select 1K samples as test set (if the original test data size exceeds 1K) and 2K samples for training (if the original training data has more than 2K samples). Given that there is no official training set for TruthfulQA, we randomly select 397 instances from the original test set for training and use the rest for testing in our experiments. For BioGen, we collect 683 people's names provided by Min et al. (2023) in total. Of these names, 183 are utilized for evaluation purposes, while the remaining 500 are employed to generate both correct and incorrect claims for training LITCAB. Similarly, for the WikiGen task, we randomly select 600 entities from the FEVER dataset, each linking to a specific Wikipedia passage. Among these entities, 100 are designated for evaluation, while the remaining 500 are used for training LITCAB. Regarding QAMPARI, we randomly selected 1K samples for testing and 2K samples for training.

## C  IMPLEMENTATION DETAILS

To ensure consistency across tasks, we use in-context learning since not all LMs exhibit good zero-shot capabilities. We use 15 demonstrations for NQ, SciQ, TriviaQA, TruthfulQA, and WikiQA, and 5 for BioGen and WikiGen because their demonstrations are much longer. In the case of BioGen, we select 5 biographies from WikiBio (Lebret et al., 2016). For WikiGen, we manually gather 5 demonstrations by extracting the initial paragraphs from Wikipedia passages. For the remaining tasks, random sampling from the training set is used. The query for BioGen is formed as "*Write a paragraph for [Name]'s biography*", and for WikiGen, it is "*Write a paragraph about [Entity]*".

| Task | Metric | | GPT2-XL(1.5B) | GPT-J(6B) | LLaMA-7B | LLaMA-13B | LLaMA-30B | Llama2-7B | Llama2-13B | Vicuna-13B |
|---|---|---|---|---|---|---|---|---|---|---|
| **Phrase Level** | | | | | | | | | | |
| NQ | acc@50 | ↑ | 0.062 | 0.146 | 0.358 | 0.402 | **0.466** | 0.288 | 0.448 | 0.246 |
| | cov@50 | ↑ | 0.001 | 0.057 | 0.271 | 0.347 | **0.445** | 0.115 | 0.407 | 0.113 |
| | ECE | ↓ | **0.045** | 0.059 | 0.144 | 0.123 | 0.169 | 0.171 | 0.139 | 0.204 |
| | Brier | ↓ | **0.055** | 0.079 | 0.174 | 0.180 | 0.192 | 0.196 | 0.187 | 0.224 |
| SciQ | acc@50 | ↑ | 0.258 | 0.620 | 0.756 | 0.796 | **0.874** | 0.764 | 0.844 | 0.678 |
| | cov@90 | ↑ | 0.007 | 0.135 | 0.261 | 0.277 | **0.423** | 0.211 | 0.375 | 0.142 |
| | ECE | ↓ | **0.059** | 0.133 | 0.126 | 0.117 | 0.107 | 0.094 | 0.102 | 0.244 |
| | Brier | ↓ | **0.137** | 0.209 | 0.210 | 0.206 | 0.186 | 0.203 | 0.197 | 0.318 |
| TriviaQA | acc@50 | ↑ | 0.100 | 0.270 | 0.474 | **0.564** | 0.462 | 0.500 | 0.454 | 0.464 |
| | cov@50 | ↑ | 0.000 | 0.128 | 0.029 | **0.426** | 0.156 | 0.111 | 0.169 | 0.268 |
| | ECE | ↓ | 0.063 | 0.068 | 0.137 | 0.133 | **0.052** | 0.112 | 0.087 | 0.186 |
| | Brier | ↓ | **0.069** | 0.115 | 0.213 | 0.213 | 0.174 | 0.203 | 0.190 | 0.238 |
| **Sentence Level** | | | | | | | | | | |
| TruthfulQA | acc@50 | ↑ | 0.186 | 0.162 | 0.012 | 0.362 | 0.433 | 0.314 | 0.362 | **0.552** |
| | cov@40 | ↑ | 0.005 | 0.040 | 0.117 | 0.331 | 0.648 | 0.136 | 0.350 | **0.998** |
| | ECE | ↓ | **0.041** | 0.112 | 0.120 | 0.121 | 0.110 | 0.138 | 0.132 | 0.200 |
| | Brier | ↓ | **0.118** | 0.136 | 0.184 | 0.223 | 0.235 | 0.218 | 0.233 | 0.303 |
| WikiQA | acc@50 | ↑ | 0.099 | 0.240 | 0.322 | 0.347 | 0.264 | 0.388 | **0.455** | 0.421 |
| | cov@50 | ↑ | 0.000 | 0.000 | 0.086 | 0.000 | 0.078 | 0.012 | **0.358** | 0.053 |
| | ECE | ↓ | 0.063 | **0.045** | 0.108 | 0.114 | 0.142 | 0.075 | 0.064 | 0.211 |
| | Brier | ↓ | **0.125** | 0.149 | 0.190 | 0.223 | 0.182 | 0.212 | 0.192 | 0.304 |
| **Paragraph Level** | | | | | | | | | | |
| BioGen | acc@50 | ↑ | – | 0.228 | 0.220 | 0.275 | 0.313 | 0.347 | **0.485** | 0.380 |
| | cov@40 | ↑ | – | 0.023 | 0.134 | 0.250 | 0.300 | 0.066 | **0.999** | 0.451 |
| | ECE | ↓ | – | 0.159 | 0.143 | 0.128 | **0.105** | 0.169 | 0.114 | 0.229 |
| | Brier | ↓ | – | 0.182 | **0.153** | 0.162 | 0.173 | 0.269 | 0.260 | 0.255 |
| WikiGen | acc@50 | ↑ | – | 0.395 | 0.667 | 0.773 | 0.750 | 0.876 | **0.900** | 0.822 |
| | cov@80 | ↑ | – | 0.001 | 0.220 | 0.450 | 0.354 | 0.745 | **0.914** | 0.675 |
| | ECE | ↓ | – | 0.102 | 0.102 | 0.124 | 0.165 | **0.045** | 0.048 | 0.168 |
| | Brier | ↓ | – | 0.220 | 0.239 | 0.226 | 0.252 | 0.172 | **0.164** | 0.227 |
| QAMPARI | acc@50 | ↑ | – | 0.115 | 0.229 | 0.293 | 0.267 | 0.193 | 0.253 | **0.305** |
| | cov@40 | ↑ | – | 0.000 | 0.387 | **0.580** | 0.501 | 0.260 | 0.464 | 0.501 |
| | ECE | ↓ | – | 0.372 | **0.257** | 0.291 | 0.268 | 0.290 | 0.323 | 0.268 |
| | Brier | ↓ | – | 0.306 | **0.228** | 0.250 | 0.230 | 0.228 | 0.265 | 0.230 |

Table 4: Different calibration metrics of popular LMs computed over CAT.

Please note that we use LITCAB exclusively to evaluate the confidence of predictions made by the original LM. LITCAB does not affect the generation process of the LM.

For searching the optimal hyperparameter of LITCAB and comparison methods, we use 20% of the training samples for validation and train LITCAB on the remaining samples. We use a training batch size of 128 and a learning rate of 1e-5. We train LITCAB for 50 epochs with early stopping. To prevent excessive adjustment of the LM's predicted logits, we initialize LITCAB's weights to zero. All LMs run on a single GPU with 48GB. We use fp16 for 13B-sized models. For models with more than 13 billion parameters, we apply a quantization technique, called GPT-Q algorithm (Frantar et al., 2022), it also enables mixed precision of int4 and float16 during inference. To implement the over-smoothing comparison method, we train the entire LLM, set the LoRA rank to 8 with a learning rate of 3e-4, utilize cross-entropy loss with the label_smoothing attribute set to 0.1, and train the model for 10 epochs. For the self-consistency method, we consider two model generations as semantically equivalent if they entail each other. We sample 10 generations for each question and use the largest cluster of semantically equivalent generations to estimate the LM confidence.

# D   RESULTS OF LMS ON CAT

The detailed results of LMs on CAT can be found in Table 4.

