# OpenReview forum: "LitCab: Lightweight Language Model Calibration over Short- and Long-form Responses"
_ICLR.cc/2024/Conference — ICLR 2024 poster_

### Official Review · Reviewer_vYMP · 2023-10-27

**Soundness:** 3 good
**Presentation:** 4 excellent
**Contribution:** 3 good
**Rating:** 8
**Confidence:** 3

**Summary:**

This paper present LITCAB, a lightweight calibration mechanism to calibrate the LLMs by only adding <2% of the original model parameters. This paper also introduces a new collection of datasets for assessing calibration of LLMs in phrase-, sentence-, and paragraph-level. Experimental results show that the proposed method outperforms a series of baselines by using four metrics. Other key findings include: (i) Larger models within the same family exhibit better calibration. (ii) GPT-family models show superior calibration compared to LLaMA, Llama-2 and Vicuna models despite having much fewer parameters. (iii) Fine-tuning pretrained model (e.g., LLaMA) with samples of focused purpose (e.g., conversations) may lead to worse calibration, highlighting the importance of fine-tuning setups.

**Strengths:**

1. The model calibration is highly meaningful, especially for nowadays LLMs, and the proposed method can be used for most of the open-source models.

2. The proposed method is simple yet effective, although it needs some training examples collected by ChatGPT and GPT-4.

3. The collected datasets CAT can also be useful for evaluating the methods for LLMs calibration.

**Weaknesses:**

1. The paper lacks discussion and comparison of some related works [1][2].

- [1] Inference-Time Intervention: Eliciting Truthful Answers from a Language Model. arXiv 2023.

- [2] Eliciting Latent Predictions from Transformers with the Tuned Lens. arXiv 2023.

2. There are still some efficient fine-tuning methods (e.g. LoRA) that can leverage additional training data, but the paper lacks results compared with such methods.

**Questions:**

1. It appears that your method involves using GPT-4 to generate training data, which has also been widely used in recent research. Does this introduce potential unfairness in the comparison? If the training data cannot be generated using GPT-4 or if its quality is poor, would the method still be effective?

2. The proposed method seems to have minimal differences compared to the original LM, and the experimental results also indicate that the original LM performs not very badly. Therefore, what are the main advantages of the proposed method? How does it improve LLMs qualitatively or quantitatively?

3. Why do you use different thresholds (e.g., cov@50, cov@90, cov@60) for different datasets?

---

> ### Author Response · Authors · 2023-11-20
> **Response to Reviewer vYMP**
>
> We truly value your constructive feedback and we've carefully addressed each of your concerns, as outlined below:
>
> ## 1. The paper lacks discussion and comparison of some related works.
>
> ITI [1] focuses on enhancing the truthfulness of LLMs and their main purpose is not strengthening the calibration performance of LLMs, so we didn’t take it as our comparison. Due to insufficient time, we only compared LitCab with ITI on TruthfulQA, which was implemented in their code. The results are shown in the table below.
>
> | Method | Original LM | ITI   | LitCab    |
> |--------|-------------|-------|-----------|
> | acc@50 | 0.314       | 0.300 | **0.314** |
> | cov@40 | 0.136       | 0.005 | **0.195** |
> | ECE    | 0.138       | 0.140 | **0.105** |
> | Brier  | 0.218       | 0.228 | **0.206** |
>
> We can see that LitCab outperforms ITI in calibration across all metrics.
>
> Tuned Lens [2] offers an analytical approach to comprehend the inference process of transformers, potentially aiding in refining models. In contrast, LitCab tackles a challenge in language models - calibration - and provides an effective, efficient solution. Tuned Lens has the potential to improve the calibration performance of language models, e.g. estimating the model confidence from the signals of Tuned Lens, and we will consider this aspect in our future work. We will carefully discuss all the work and their relation to our method in the revision.
>
> ## 2. The paper lacks results compared with efficient fine-tuning methods.
>
> In Sections 6.3 and 6.4 (highlighted in green), we introduced and compared LitCab with Label Smoothing, where the LM is fine-tuned using LoRA. Results in Table 1 indicate that it does not yield satisfactory calibration performance, possibly due to the small size of the training data leading to model overfitting. In contrast, LitCab can leverage small training sets for improved calibration, suggesting that LitCab achieves superior data efficiency compared to label smoothing.
>
> ## 3. Using GPT-4 to generate training data
>
> To clarify, we only use GPT-4 to evaluate the correctness of the LM's responses. All the incorrect responses for training LitCab are generated by the LM itself.
>
> ## 4. What are the main advantages of the proposed method? How does it improve LLMs qualitatively or quantitatively?
>
> In response to the observation of the baseline's relatively low ECE, which falls within the range of approximately 0.1 to 0.3, it is important to note that the typical ECE range is not confined to 0 to 1, as randomness in model confidence and correctness alone can yield an ECE of 0.25. However, our proposed method, LitCab, has demonstrated a significant and noteworthy impact on calibration. Table 1, highlighted in green, serves as compelling evidence, showcasing LitCab's effectiveness in achieving a remarkable 25% decrease in the average expected calibration error—from 0.118 to 0.089. This substantial reduction underscores the significant improvement LitCab brings to calibration, emphasizing its practical importance and contribution to the field.
>
> ## 5. Why do you use different thresholds (e.g., cov@50, cov@90, cov@60) for different datasets?
>
> This is because language models demonstrate varied performance across different datasets. Cov@p, as defined, assesses the model's recall by identifying the largest percentage denoted as 'c,' for which the most confident c percent of predictions exhibit accuracy surpassing the threshold 'p.' It is crucial to note that a lower or higher value of 'p' can result in meaningless comparisons; for example, all models will achieve 100% when p is set to 0 and 0% when p is set to 1.0.
>
> To illustrate, consider SciQ, which is relatively easy for LMs. If we set a lower value of 'p' in cov@p (e.g., 50), it results in all 100% for all models in cov@50. This poses a challenge in utilizing this metric for comparisons. Therefore, in alignment with the approach outlined by Tian et al. (2023) [3], we opt for a relatively high value of 'p' to facilitate more meaningful and informative comparisons.
>
> It is important to emphasize that we do not solely rely on this metric (due to its aforementioned limitations), we also report acc@q, ECE, and Brier score, to include a comprehensive set of metrics for evaluating the calibration performance of models.
>
> **Reference**
>
> [1] Inference-Time Intervention: Eliciting Truthful Answers from a Language Model. arXiv 2023.
>
> [2] Eliciting Latent Predictions from Transformers with the Tuned Lens. arXiv 2023.
>
> [3] Katherine Tian, Eric Mitchell, Allan Zhou, Archit Sharma, Rafael Rafailov, Huaxiu Yao, Chelsea Finn, and Christopher D Manning. Just ask for calibration: Strategies for eliciting calibrated confidence scores from language models fine-tuned with human feedback. Arxiv 2023.

---

> > ### Author Response · Authors · 2023-11-22
> > **Reminder**
> >
> > Dear Reviewer vYMP
> >
> > Thank you for your dedicated review efforts. As the discussion stage deadline approaches, please feel free to raise any concerns or seek clarification on our manuscript. Any feedback is greatly appreciated and will help us further enhance this work.
> >
> > Thank you,
> >
> > Authors

---

> > ### Comment · Reviewer_vYMP · 2023-11-23
> >
> > Thanks for your response. I'll keep my current ratings.

---

### Official Review · Reviewer_Y9cb · 2023-11-01

**Soundness:** 2 fair
**Presentation:** 3 good
**Contribution:** 3 good
**Rating:** 6
**Confidence:** 3

**Summary:**

The paper comprises two main contributions focusing on language model calibration. Firstly, they propose Litcab, which is a lightweight LM calibration mechanism that accepts the sentence representation as input and predicts a bias term that will be added to the output logits. Secondly, they construct CaT, which is a benchmark consisting of six open-ended question-answering tasks covering text lengths of different granularity. The paper presents experiments showing that Litcub is an effective calibration mechanism, as well as a comprehensive evaluation of LLMs in terms of how well-calibrated they are using the CaT benchmark.

**Strengths:**

* A lightweight alternative to supervised model calibration is important given that our models are increasing in size.

* Calibration benchmarks for long-form responses have not yet been introduced before.

**Weaknesses:**

* Presentation-wise, parts of the paper have coherence issues because it is convoluted with two different ideas that are not properly tied up together. The paper starts to be read as having Litcab as the major contribution and curating the CaT benchmark, a minor contribution, to evaluate Litcab. However, the final page (Section 6.6 and Table 3) is dedicated to using CaT to benchmark how well-calibrated the LLMs are. Litcab is not used in this section, which makes it somewhat irrelevant to answering the main hypothesis of the paper.

* The CaT benchmark has two major shortcomings. Firstly, the motivation for creating CaT is to be able to evaluate calibration for long-form responses. However, CaT consists of five datasets with short-form responses and only one with long-form responses. There are a couple of datasets that could have been included, such as ELI5, WebGPT, ASQA, and QAMPARI (which is not passage-level but contains multiple claims). Secondly, it is important for a proposed benchmark to have both proposed (a) baselines and (b) evaluation metrics. However, the proposed baselines could not be evaluated in all cases (specifically passage-level) using their metrics. This makes it difficult to assess the calibration capability of Litcab at the passage level.

* I think it would help the paper if there were extrinsic evaluations to support Litcab's usage in a real-world task setup. Self-consistency is now commonly used because the original paper showed promising results on arithmetic and commonsense reasoning benchmarks. How would the use of Litcab fare on those datasets?

* The paper mentions that Litcab has the "advantage of requiring a reduced number of data samples for training, leading to enhanced computational and data efficiency". There is no evidence shown in the paper that supports this statement.

**Questions:**

* How would the use of Litcab fare on arithmetic and commonsense reasoning benchmarks?

---

> ### Author Response · Authors · 2023-11-20
> **Response to Reviewer Y9cb Question 1**
>
> We sincerely appreciate your constructive feedback, and we have addressed each of your concerns as outlined below:
>
> ## 1. Coherence issues.
> In this work, we present four primary contributions. Firstly, we introduce LitCab, a lightweight calibration mechanism. Its effectiveness is demonstrated through comparisons with various competitive approaches. Secondly, we introduce CaT, designed to assess text generation tasks with responses of varying lengths. Thirdly, we extensively evaluate the calibration of seven state-of-the-art open-source language models on CaT, deriving three key findings related to model size, model family, and model fine-tuning. Fourthly, we formulate an evaluation strategy for paragraph-level model generations. In the revised introduction, we have highlighted the significance of the third contribution, found in the second paragraph from the bottom, which is marked in blue.

---

> ### Author Response · Authors · 2023-11-20
> **Response to Reviewer Y9cb Question 2**
>
> ## 2. There is only 1 dataset with long-form responses. More datasets, such as ELI5, WebGPT, ASQA, and QAMPARI should be considered.
>
> Thank you for your suggestion. We followed your advice and conducted the experiment on QAMPAR because there are several issues when experimenting with other datasets. Regarding ELI5, collecting support documents for correctness evaluation is extremely time-consuming, and unfortunately, we won't be able to do that due to the limited rebuttal time. However, we will definitely consider experimenting with ELI5  in the future. As for WebGPT, there is currently no available and consistent source for checking the correctness of model responses, given the open-ended nature of the questions. While quoted sentences are provided, they are not sufficient for verifying the knowledge in various model responses. ASQA primarily inspects the model's capability on ambiguous questions, which includes the uncertainty of questions, and it's not within our scope. This leaves QAMPARI, where the correctness of model responses can be verified by comparing them with provided references. LitCab demonstrates its superiority in this task, based on the results below
> |   Task  | Metric | Original LM | Label Smoothing | Temp. Scaling | LitCab    | LitCab w/ Temp.Scaling |
> |:-------:|--------|-------------|-----------------|---------------|-----------|------------------------|
> | QAMPARI | acc@50 | 0.193       | 0.180           | 0.188         | **0.207** | **0.207**              |
> |         | cov@30 | **0.260**   | 0.156           | 0.239         | 0.257     | 0.257                  |
> |         | ECE    | 0.290       | 0.213           | 0.303         | **0.096** | 0.104                  |
> |         | Brier  | 0.228       | 0.208           | 0.273         | **0.142** | 0.157                  |
>
> Results of LitCab and baselines on CaT
>
> Furthermore, to evaluate LitCan on more tasks, we introduce a new task for long-form generation—WikiGen, where the LM is tasked to generate Wiki-style descriptions for entities, with an average generation length of 104, sourced from the fact verification dataset FEVER [1]. Compared with BioGen which is limited to biographies, WikiGen covers a wider spectrum of subjects beyond individuals, including events, concepts, and objects, resulting in more diverse topics. LitCab again obtains the best calibration performance, as evidenced by the results presented in the following table:
>
> | Task    | Metric | Original LM | Label Smoothing | Temp. Scaling | LitCab    | LitCab w/ Temp.Scaling |
> |---------|--------|-------------|-----------------|---------------|-----------|------------------------|
> | WikiGen | acc@50 | 0.876       | 0.860           | 0.876         | **0.872** | **0.872**              |
> |         | cov@80 | 0.745       | 0.733           | 0.745         | **0.756** | **0.756**              |
> |         | ECE    | 0.045       | 0.075           | 0.049         | **0.037** | 0.065                  |
> |         | Brier  | 0.172       | 0.187           | 0.173         | **0.171** | 0.174                  |
>
> Results of LitCab and baselines on CaT
>
> We also evaluate seven LLMs using both QAMPARI and WikiGen:
>
> | Task    | Metric | GPT-J (6B) | LLaMA-7B  | LLaMA-13B | LLaMA-30B | Llama2-7B | Llama2-13B | Vicuna-13B |
> |---------|--------|------------|-----------|-----------|-----------|-----------|------------|------------|
> | QAMPARI | acc@50 | 0.115      | 0.229     | 0.293     | 0.267     | 0.193     | 0.253      | **0.305**  |
> |         | cov@30 | 0.000      | 0.387     | **0.580** | 0.501     | 0.260     | 0.464      | 0.501      |
> |         | ECE    | 0.372      | **0.257** | 0.291     | 0.268     | 0.290     | 0.323      | 0.268      |
> |         | Brier  | 0.306      | **0.228** | 0.250     | 0.230     | **0.228** | 0.265      | 0.230      |
>
> | Task    | Metric | GPT-J (6B) | LLaMA-7B | LLaMA-13B | LLaMA-30B | Llama2-7B | Llama2-13B | Vicuna-13B |
> |---------|--------|------------|----------|-----------|-----------|-----------|------------|------------|
> | WikiGen | acc@50 | 0.395      | 0.667    | 0.773     | 0.750     | 0.876     | **0.900**  | 0.822      |
> |         | cov@80 | 0.001      | 0.220    | 0.450     | 0.354     | 0.745     | **0.914**  | 0.675      |
> |         | ECE    | 0.102      | 0.102    | 0.124     | 0.165     | **0.045** | 0.048      | 0.168      |
> |         | Brier  | 0.220      | 0.239    | 0.226     | 0.252     | 0.172     | **0.164**  | 0.227      |
>
> Different calibration metrics of popular LMs computed over CaT
>
>
> **Reference**
>
> [1] James Thorne, Andreas Vlachos, Christos Christodoulopoulos, and Arpit Mittal. 2018. FEVER: a Large-scale Dataset for Fact Extraction and VERification. NAACL 2018.

---

> ### Author Response · Authors · 2023-11-20
> **Response to Reviewer Y9cb Question 3,4,5**
>
> ## 3. The proposed baselines could not be evaluated in all cases.
>
> To provide comparisons, we have to create training samples for Label Smoothing and Temporal Scaling in paragraph-level tasks where ground truth is unavailable. This involves gathering accurately generated claims, assessed using GPT-4.
>
> We present the results of Label Smoothing and Temporal Scaling on paragraph-level tasks in Table 1 in our revised version (highlighted in blue). The results suggest that both methods adversely affect the LM's calibration performance. This is attributed to tuning the temperature or fine-tuning the entire LLM based on generated samples leading to overconfidence.
>
>
> ## 4. How would the use of Litcab fare on arithmetic and commonsense reasoning benchmarks?
>
> In this paper, we focus on evaluating the calibration of LMs on outputs of varied lengths. Therefore, we design LitCab in the context of generation tasks and conduct experiments related to such tasks.
>
> Following your suggestion, we further conduct experiments on the CommonsenseQA task under the multiple-choice setting. We compare the results of the original LM, self-consistency and LitCab. LitCab demonstrates superior calibration performance compared to both the original language model and self-consistency, as evidenced by its lower ECE and Brier score, as outlined below:
>
> | Task                            | Metric | Original LM | Self-consistency | LitCab |
> |---------------------------------|--------|-------------|------------------|--------|
> | CommonsenseQA (multiple-choice) | acc@50 | 0.237       | 0.243            | 0.235  |
> |                                 | cov@60 | 0.006       | 0.011            | 0.011  |
> |                                 | ECE    | 0.039       | 0.196            | 0.034  |
> |                                 | Brier  | 0.166       | 0.209            | 0.165  |
>
> ## 5. There is no evidence shown in the paper that supports "advantage of requiring a reduced number of data samples for training, leading to enhanced computational and data efficiency".
>
> We apologize for our imprecise claim. We have rephrased the corresponding description in the revision: LitCab achieves superior computational efficiency with fewer tunable parameters, compared with the fine-tuning-based baselines.

---

> ### Author Response · Authors · 2023-11-20
> **Response to Reviewer Y9cb**
>
> We hope our explanations and modifications have addressed your concerns. If you find them satisfactory, we kindly ask you to take these improvements into consideration when reassessing our work. We are open and would greatly appreciate any additional comments, follow-up questions, or suggestions related to our work.

---

> ### Author Response · Authors · 2023-11-22
> **Reminder**
>
> Dear Reviewer Y9cb
>
> Thank you for your dedicated review efforts. As the discussion stage deadline approaches, please feel free to raise any concerns or seek clarification on our manuscript. Any feedback is greatly appreciated and will help us further enhance this work.
>
> Thank you,
>
> Authors

---

> > ### Comment · Reviewer_Y9cb · 2023-12-01
> > **Thanks**
> >
> > Thanks for the response, I have raised the score a bit.

---

### Official Review · Reviewer_7atx · 2023-11-02

**Soundness:** 3 good
**Presentation:** 3 good
**Contribution:** 2 fair
**Rating:** 5
**Confidence:** 3

**Summary:**

The paper proposes a new calibration technique for LLMs: LITCAB, a lightweight calibration mechanism that adjusts the generation confidence of large language models (LLMs) by adding and training a single linear layer over the last hidden states of the LLM. LITCAB aims to improve the alignment between the model confidence and the output correctness and reduce the issue of hallucination in LLMs. The authors construct CAT, a calibration evaluation benchmark consisting of six open-ended question answering (QA) tasks that cover answers ranging from phrases to paragraphs. The authors propose a four-step procedure to assess the calibration and correctness of paragraph-level generations, which involves extracting individual claims, mapping them to spans, estimating their confidence and verifying their accuracy using GPT-4 and Wikipedia passages. The authors conduct experiments on LITCAB and several baselines using Llama2-7B and other popular open-source LLMs. They find that LITCAB consistently improves the calibration performance across all tasks, and that larger models within the same family tend to exhibit better calibration. They also observe that fine-tuning may lead to worse calibration in some cases.

**Strengths:**

- Originality: The authors propose LITCAB, a lightweight calibration mechanism that only adds and trains a single linear layer on top of the LLMs12. The authors also construct CAT, a new benchmark for evaluating calibration across different answer lengths.
- Quality: The paper compares several baselines of calibration on datasets of different output length - phrase, sentence, and paragraph.
- Clarity: The paper is easy to follow and understand.
- Significance: The paper addresses the calibration problem for LLM and improves upon SOTA open-sourced LLMs on open-ended QA tasks.

**Weaknesses:**

- It is not clear to me how to select the incorrect answers. This could be important as different selection strategies may affect the performance of the final model.
- It is not clear why adding a linear layer on the last hidden state is the best option. Other options include LoRA or prefix tunning, which also only requires a fraction of parameters to tune. More evaluation results should be presented to justify the model choice of LITCAB.

**Questions:**

- How did you select/generate incorrect answers for the QA tasks?
- The confidence of a sentence is the geometric mean of all its tokens. However, different tokens contribute differently to semantics. For instance, think about "The 2023 Oscar best actress is Michelle Yeoh" as the answer to "Who wins the 2023 Oscar best actress?". Only the last two tokens decide the correctness of the sentence. How did you handle the different importance of tokens?

---

> ### Author Response · Authors · 2023-11-20
> **Response to Reviewer 7atx**
>
> Thank you for your thoughtful comments and suggestions. We have carefully considered all of them and addressed each one in the revised version of our paper. We truly believe that these modifications and clarifications significantly improved our work.
>
> ## 1. It’s not clear how to select the incorrect answers.
>
> For phrase- and sentence-level tasks where references are available, we use the references directly as positive samples. We then repeatedly sample model generations, treating incorrect ones, evaluated by GPT4, as negative samples. For the paragraph-level tasks, we directly instruct each LM to generate biographies or descriptions for the given people’s names or entities. Subsequently, we categorize the correct claims as positive samples and the incorrect ones as negative samples. We have included this description in the final paragraph of section 4 in our revised version–highlighted in blue.
>
> ## 2. It’s not clear why adding a linear layer on the last-layer hidden states is the best option.
>
> This is because we aim to prioritize computational efficiency in the design of the lightweight calibration mechanism. Fine-tuning the LM through LoRA involves gradient back-propagation through all layers of LM. In contrast, training LitCab only necessitates running the LM's inference process to acquire the last-layer hidden states and logits. The single linear layer can then be trained based on these gathered features.
>
> In our pilot study, we experimented with both the second-to-last layer and the last layer's hidden states and found that using the last layer performed better. This is likely since because the last layer is closest to the logits and encodes more information about the future token. Exploring the optimal layer or combinations of the layers is a topic we intend to delve into in our future work.
>
> In Section 6.3 and 6.4 (highlighted in green), we compare LitCab with Label Smoothing fine-tuning with LoRA. Results in Table 1 indicate that Label Smoothing does yield satisfactory calibration performance, possibly due to the small size of the training data leading to model overfitting. In contrast, LitCab can leverage small training sets for improved calibration, suggesting that LitCab achieves superior data efficiency compared to label smoothing.
>
> ## 3. How to handle the different importance of tokens?
>
> We answer your comment in two points. Firstly, this aggregation method has been studied and shown as effective in approximating the sentence-level probability by Kuhn et al., 2023 [1], and Si et al., 2023 [2], we aggregate token probabilities to calculate the sentence probability in our work.
>
> Secondly, we argue that there is no straightforward definition of what constitutes an important token, as **the token importance heavily relies on the context**, such as the question, and it could easily change if any tokens related to the question are altered. In reference to your example, the model may generate an answer like "The 2022 Oscar best actress is Michelle Yeoh." If we only focus on the individual’s name, the incorrect generation of "2022" might be easily overlooked.
>
> In light of these considerations, we chose to treat all tokens equally for the sake of simplicity and generalizability to text of varied lengths in this work. We are also open to and appreciate any suggestions from this perspective.
>
> **Reference**
>
> [1] Kuhn, Lorenz, Yarin Gal, and Sebastian Farquhar. Semantic uncertainty: Linguistic invariances for uncertainty estimation in natural language generation. ICLR 2023.
>
> [2] Si, Chenglei, Zhe Gan, Zhengyuan Yang, Shuohang Wang, Jianfeng Wang, Jordan Boyd-Graber, and Lijuan Wang. Prompting gpt-3 to be reliable. ICLR 2023.

---

> ### Author Response · Authors · 2023-11-20
> **Response to Reviewer 7atx**
>
> We hope that these clarifications address your concerns. We believe the paper has been significantly improved thanks to your comments.
>
> If any points remain unclear, or if you have additional questions or suggestions, please feel free to share them.
>
> In light of these improvements and clarifications, we kindly request that you consider adjusting your scores.
>
> Thank you once again for your time and effort in reviewing our work. Your valuable insights are greatly appreciated.

---

> ### Author Response · Authors · 2023-11-22
> **Reminder**
>
> Dear Reviewer 7atx
>
> Thank you for your dedicated review efforts. As the discussion stage deadline approaches, please feel free to raise any concerns or seek clarification on our manuscript. Any feedback is greatly appreciated and will help us further enhance this work.
>
> Thank you,
>
> Authors

---

### Meta-Review · Area_Chair_ToDK · 2023-12-09

**Metareview:**

In this paper, the authors presented LITCAB, a lightweight calibration technique for LLMs. LITCAB effectively calibrates LLMs by predicting a bias term added to the model’s predicted logits. The extensive experiments substantiate the efficacy of LITCAB and underscore its computational efficiency. Moreover, the authors collected CAT, a benchmark specifically designed to evaluate calibration in QA settings with answers of varying lengths. We use CAT to comprehensively assess seven open-source LLMs. Finally, the authors proposed an evaluation methodology dedicated to examining calibration in long-form generations, which can serve as the basis for future studies.

The paper is well-written and easy to follow. The idea is interesting and novel. During the rebuttal, the reviewers asked several questions about experiments, motivation, and some technical details of the paper and the authors successfully addressed all of them.

**Justification For Why Not Higher Score:**

The technical contribution of the paper is still a bit incremental

**Justification For Why Not Lower Score:**

N/A

---

### Decision · Program_Chairs · 2024-01-16

Accept (poster)